# Relationships between Weight Perceptions and Suicidal Behaviors in Chinese Adolescents: Results from an Ongoing School-Based Survey in Zhejiang Province

**DOI:** 10.3390/bs13010008

**Published:** 2022-12-22

**Authors:** Zhu Yu, Fan He, Gaofeng Cai, Meng Wang, Junfen Fu

**Affiliations:** 1Department of Endocrinology, Children’s Hospital Zhejiang University School of Medicine, 3333 Binsheng Road, Hangzhou 310051, China; 2National Clinical Research Center for Child Health, 3333 Binsheng Road, Hangzhou 310051, China; 3National Children’s Regional Medical Center, 3333 Binsheng Road, Hangzhou 310051, China; 4Zhejiang Provincial Center for Disease Control and Prevention, 3399 Binsheng Road, Hangzhou 310051, China

**Keywords:** weight perception, adolescents, suicide, relationships, China

## Abstract

Background: Weight perception has been implicated in suicidal behaviors among children and adolescents, yet little is known about the relationships in China. We examined the associations of self-perceived weight status and weight misperception with suicidal behaviors among Chinese adolescents. Methods: Data used in this analysis were from the 2017 Zhejiang Youth Risk Behavior Survey, which included 17,359 middle and high school students aged 9 to 18 years. Information on perceived weight status, as well as the actual height, weight and other related traits, was extracted by a self-reported questionnaire. Multivariate logistic regression models were used to estimate adjusted odds ratios (ORs) with 95% confidence intervals (CIs) for suicidal behaviors associated with perceived weight status and weight misperception. Results: Overall, the mean (SD) age was 15.7 (1.6) years among the study participants. Students who perceived themselves as overweight were more likely to have increased suicidal ideation and attempts reports, with ORs of 1.22 (95% CI: 1.09–1.37) and 1.32 (1.06–1.34), compared to those who perceive themselves as having about the right weight. Overestimated weight was significantly associated with greater odds of suicidal ideation (OR: 1.15, 95% CI: 1.03–1.27) and attempts (1.35, 1.10–1.65) relative to accurate weight perception. Conclusions: Controlling for potential confounders, we found that both self-perception of overweight and overestimated perception were significantly associated with suicidal ideation and attempts among Chinese adolescents.

## 1. Introduction

Obesity in children and adolescents has emerged as one of the most serious public health concerns worldwide. Since 1975, the prevalence of obesity among boys and girls has increased remarkably, with obese individuals exceeding 100 million in 2016 [1]. Obesity in children and adolescents can increase the risk of negative physical outcomes, such as hypertension, cardiovascular diseases and type 2 diabetes [2,3,4]. Besides the effects on physiologic health, obesity in children and adolescents can also lead to adverse mental-health-related consequences, including depression, anxiety, low self-esteem, attention-deficit hyperactivity disorder and suicide [5,6].

Among these psychosocial effects of obesity, due to the tremendous disease burden, suicide problems draw much public health attention. According to the latest estimates from the World Health Organization in 2019, more than 700,000 persons died by suicide every year globally, and suicide has been the second-leading cause of death among 10–24-year-old youths [7]. Furthermore, it is noteworthy that, as the immediate precursors to suicide, nonfatal suicidal behaviors, including suicidal ideation, plans and attempts, are also a severe public health problem, with overall prevalence of 16.5%, 16.5% and 16.4%, receptively, among adolescents aged 12–15 years from 83 countries [8]. Previous studies found that various factors were related to nonfatal suicidal behaviors, including sex, race/ethnicity, academic achievement, smoking and depression [9,10,11]. Recently, researchers have taken an increasing interest in the possible impacts of overweight and obesity on nonfatal suicidal behaviors in youths. Both cross-sectional and longitudinal studies suggested that overweight and obesity were associated with suicidal ideation and attempts, while these associations might be mediated by some factors, such as sex [12,13,14], victimization [15], academic pressure and smoking [12], in diverse cultures and countries. Furthermore, there are some observational findings that the detrimental effect of overweight and obesity on suicidality is largely dependent on the self-perception of overweight and obesity rather than one’s objective weight status. For example, a systematic review and meta-analysis study observed that a perception of overweight was positively associated with suicidality independently, and the association between objective higher weight status and suicidality was attenuated to non-significance after adjusting for perceived overweight [16]. Similarly, in a nationally representative cohort study of Chinese adolescents, the authors found that perceiving oneself as overweight or obese was significantly related to suicidal behaviors, while the actual measured weight was not [17]. Adolescence is the transition from childhood to adulthood with many psychological changes taking place. Weight perception, which involves the identification and estimation of body weight, is a primary dimension of body image central to emotional well-being [18]. Based on the national surveys conducted in US, South Korea and China, about a quarter of children and adolescents have perceived themselves to be overweight or obese [17,19,20], while nearly half inaccurately estimated their weight status [21,22,23]. Considering the high prevalence of perceived overweight or obese and weight misperception, and the huge challenge of suicide in children and adolescents, understanding the relationships of weight perceptions with these nonfatal suicidal behaviors is helpful in the development of more early public health interventions on suicide prevention.

Therefore, according to the literature review, we proposed the hypothesis that weight perceptions were related to suicidality, and this study was intended to explore the possible associations of self-perceived weight status and weight misperception with suicidal behaviors, including suicidal ideation, plans and attempts, in Chinese adolescents.

## 2. Materials and Methods

### 2.1. Design and Participants

Participants included in this analysis were from the Zhejiang Youth Risk Behavior Survey (YRBS), which is conducted by the Zhejiang Provincial Center for Disease Control and Prevention (CDC). The YRBS study is an ongoing school-based study designed to assess the prevalence of health-related behaviors and associated factors among Chinese adolescents. The study design and participants’ characteristics have been previously described in details elsewhere [24]. Briefly, data used in the present analysis were from the latest survey conducted in 2017, in which multistage, stratified cluster sampling methods were adopted, with 23,554 middle and high school students were successfully recruited. After excluding the ineligible questionnaires and participants with missing key information such as age, gender and height and weight, 17,359 students were finally included in this analysis. Information on demographic characteristics and health-related behaviors, including smoking, alcohol drinking, dietary intake, physical activity, sleep duration, weight control attempt and strategies, and suicidal behaviors, was collected via a self-administered questionnaire derived from the US 1991–2015 Youth Risk Behavior Surveillance System (YRBSS) and the Global School-based Student Health Survey (GSHS). The students completed the anonymous questionnaire in the classroom independently, and afterward, questionnaires were collected by the researchers. To make all participation voluntary, parents/guardians of the students and the school officials were sent a written letter and given the option to refuse. Verbal consent was obtained from parents/guardians of the selected students to publish the collected data without any identity information. Additionally, all the researchers were strictly trained to protect the students’ privacy and ensure the confidentiality of the personal data. In particular, this study abided by the “Declaration of Helsinki” and was approved by the ethics committee of Zhejiang CDC.

### 2.2. Assessment of Weight Perceptions

In this study, self-perceived weight status was assessed by the following question, “How do you describe your weight?”, with response options as follows: very underweight, slightly underweight, about the right weight, slightly overweight and very overweight. Participants who reported being “very” or “slightly” underweight were categorized as “underweight” and those who reported being “very” or “slightly” overweight were categorized as “overweight”.

The accuracy of weight perceptions was classified as overestimated, underestimated or accurate by comparing the self-perceived and the actual weight status based on body mass index (BMI, underweight, normal weight, overweight, obesity). BMI was categorized using the age-and sex-specific cut-off points developed by the Childhood Obesity Working Group of the International Obesity Task Force (IOTF) [25,26]. To make the comparisons between the self-perceived and actual weight status, we combined the specific BMI groups of overweight and obesity into a single category (overweight/obesity). Overestimated perception included those underweight adolescents who perceived themselves as about the right weight or overweight and normal weight adolescents who perceived as overweight. Underestimated perception included normal weight adolescents who perceived themselves as underweight and overweight/obesity adolescents who perceived themselves as about the right weight or underweight. Neither overestimated nor underestimated perception was defined as accurate weight perception.

### 2.3. Assessment of Suicidal Behaviors

Suicidal ideation, plans and attempts were assessed with the following questions, respectively, “Have you seriously considered suicide in the last 12 months?”, “Have you made a specific plan to commit suicide in the last 12 months?” and “Have you attempted suicide in the last 12 months?” The response options to the above questions were “yes” or “no”.

### 2.4. Other Covariates

Socio-demographic characteristics, some lifestyle behaviors and other mental-health-related characteristics potentially associated with suicidal behaviors, were also considered in this analysis. They included age (≤13, 14, 15, ≥16 years), gender (boys, girls), location of school (rural, urban), school type (middle school, academic high school, vocational high school), school performance (above average, average, below average), paternal education (middle school or below, high school, college or above), maternal education (middle school or below, high school, college or above), BMI (underweight, normal weight, overweight/obesity). Besides, current smoking, current alcohol drinking, muscle strengthening activity, other mental health-related characteristics of feelings of loneliness, sleep loss due to worry, experience of sadness/despair and weight control related behaviors of exercising, dieting, taking laxatives, taking diet pills and fasting were also considered.

### 2.5. Statistical Analysis

Descriptive statistics were used to describe the characteristics of students according to suicidal behaviors including suicidal ideation, plans, and attempts. The difference in characteristics of students with or without any suicidal behavior was compared using the Kruskal–Wallis test for continuous variables and linear-by-linear association χ^2^-test for categorical variables. Multivariate logistic regression models were used to estimate the odds ratios (ORs) with 95% confidence intervals (CIs) for the associations of self-perceived weight status and the accuracy of weight perceptions with suicidal behaviors. Adjustments for confounding factors were conducted in four sequential models. In model 1, socio-demographic characteristics of age, gender, location of school, school type, school performance, paternal education and maternal education were adjusted. Model 2 was further adjusted for lifestyle behaviors of smoking, alcohol drinking, physical activity and physical measurement of BMI. Model 3 was adjusted for all variables in model 2 plus mental-health-related characteristics of feelings of loneliness, sleep loss due to worry and experience of sadness/despair. Model 4 was adjusted for all variables in model 3 plus weight-control-related behaviors of exercising, dieting, taking laxatives, taking diet pills and fasting. On the basis of model 4, associations between self-perceived weight status, the accuracy of weight perceptions and suicidal behaviors were compared within subgroups defined by age, gender, location of school, current smoking, current alcohol drinking and BMI. Additionally, to evaluate the robustness of our estimates, sensitivity analysis was conducted, excluding adolescents with current smoking and current alcohol drinking behaviors in the study. All analyses were performed using SAS statistical package (version 9.4, SAS Institute, Inc., Cary, NC, USA). All statistical tests were based on a two-sided 5% level of significance.

## 3. Results

### 3.1. Participants Characteristics

Among the participants (*n* = 17,359) in this study, the mean (SD) age was 15.7 (1.6) years, with age ranging from 9 to 18 years. Their other characteristics were shown in Table 1. Nearly half of the students were boys (49.6%) and middle school students (51.3%). The majority of students were from a rural school (60.9%) and reported that their parents had a middle school or below education level (paternal, 60.2%, and maternal, 65.4%). About a fourth of the students reported their school performance as being below average (24.9%). Few students were current smokers (5.5%) or overweight/obese (8.8%) or had weight control behaviors, such as dieting (3.7%), taking laxatives (2.0%) and taking diet pills (2.9%). Overall, 34.2% of students perceived themselves as overweight and the percentage of overestimated perception was 35.5%. In the last 12 months, 16.3% of students reported considering suicide, 6.3% planning for suicide, and 3.7% attempting suicide. Compared to students without any suicidal behavior, those with suicidal ideation or plans or attempts tended to be younger; girls; from urban schools; middle school students; with below average performance; low maternal education; and a higher proportion of smoking, alcohol drinking, mental-health-related characteristics, self-perceived overweight, overestimated perception and weight-control behaviors, while having a lower likelihood of participating in muscle-strengthening activity.

### 3.2. Associations of Self-Perceived Weight Status and Weight Misperception with Suicidal Behaviors

After adjustment for potential confounders, including socio-demographic characteristics, lifestyle behaviors, mental-health-related characteristics and weight control behaviors, the self-perceived overweight status and overestimated perception were significantly associated with suicidal behaviors of ideation and attempts but not for plans. Specifically, compared with those perceiving themselves as having about the right weight status, students who perceived themselves as overweight were significantly more likely to report suicidal ideation and attempts, with ORs of 1.22 (95% CI: 1.09–1.37) and 1.32 (1.06–1.34), while for suicidal plans, the OR was 1.05 (0.89–1.25). Relative to those with accurate weight perception, students who overestimated their weight status had significantly increased odds of reporting suicidal ideation and attempts, with ORs of 1.15 (1.03–1.27) and 1.35 (1.10–1.65), while for suicidal plans, the OR was 1.13 (0.97–1.32). Furthermore, in this analysis, neither self-perceived underweight status nor underestimated perception was associated with any suicidal behaviors (Table 2). No heterogeneity was observed for the associations of self-perceived weight status and weight misperception with suicidal ideation, plans and attempts by location of school, current smoking and BMI, although these associations differed statistically as regards age, gender and current alcohol drinking (all *p* for heterogeneity <0.05) (Appendix A). In the sensitivity analysis, although the ORs varied slightly, the associations of self-perceived weight status and weight misperception with suicidal behaviors were broadly consistent among the subsets of students without smoking and alcohol drinking (Appendix A).

## 4. Discussion

Although previous studies have indicated that extreme weight control behaviors, substance use and eating disorders were associated with suicidality [27,28,29], the relevant weight perceptions are less studied as factors related to suicidal behaviors among children and adolescents, particularly in China, with different socioeconomic and culture characteristics. Based on 17,359 Chinese middle and high school students from Zhejiang Province, the present study suggested that there were significant associations of perceived weight status and weight misperception with suicidal behaviors. Specifically, in Chinese students, we found that both self-perception of overweight and overestimated perception were associated with higher odds of suicidal ideation and attempts after controlling for potential confounders. Notably, in the subgroup analysis by gender, the significant associations of the aforementioned weight perceptions with suicidal ideation only existed in girls. Meanwhile, we observed that neither self-perceived underweight nor underestimated perception was linked to suicidal behaviors of ideation, plans or attempts.

There is mechanistic evidence that suicidal behaviors are influenced by psychological consequences of depression, distress and low self-esteem, which are relevant to body weight perception and weight misperception [16,30,31]. However, to date it is not clear what role specific self-perceived weight status and weight misperception play in the prevalence of suicidal behaviors among children and adolescents. For one thing, regarding the self-perception of weight status, data from the US 2015 Youth Risk Behavior Survey of 9th through 12th grade students showed that both overweight and underweight perceptions were significantly associated with considered suicide and attempted suicide, while only overweight perception was associated with making suicide plans [19]. Similarly, a national cross-sectional study used data from the 2015–2017 Korea Youth Risk Behavior Web-Based Surveys (KYRBWS), suggesting that adolescents with the perceptions of being overweight and underweight were linked to increased reports of suicidal ideation and planning [32]. Furthermore, in a longitudinal study with a representative sample of US middle and high school students, the authors reported that underweight perception did not have a significant effect on suicidal ideation in boys and girls [33]. By contrast, our study confirmed the positive associations of overweight perception with suicidal ideation and attempts in literature and provided more supports for the idea that the perception of underweight might not be related to suicidal behaviors. Notably, the sex differences in the relationships between perceived weight status and suicidal behaviors in adolescents have been noticed in published reports but still with inconsistent findings. For example in the United States, one study of middle school students from North Carolina provided findings that females who perceived themselves as overweight were more likely to have suicidal thoughts and actions, while for males, the significant associations were observed among those with perceptions of overweight and underweight [34]. Another US study using data from the 1999–2007 Youth Risk Behavioral Surveillance System revealed that the perception of being overweight was significantly associated with suicidal behaviors of ideation and attempts for girls, but the results were insignificant for boys [35]. Differently in China, a nationally school-based cohort study proposed that the self-perceptions of obesity, as well as overweight and underweight, exhibited significant adverse effects on suicidal behaviors among male adolescents but not among female adolescents [17]. Specific to the present study with subgroup analysis by gender, our results showed that overweight perception was significantly associated with increased suicidal ideation reports only in girls, which to some extent, was consistent with findings in a prospective study of US adolescents [33]. For another, despite the high prevalence of weight misperception among adolescents across countries, there has been limited focus on the linkage with suicidal behaviors, except for in Korea. Specifically, based on the 2017 KYRWBS of 62,276 middle and high school students, the authors reported that adolescents who overestimated their body weight were more likely to experience suicidal ideation and planning [36]. Among 20,264 normal-weight high school students selected from the 2014 KYRWBS, data indicated that adolescents with overestimation of body weight status were significantly more likely to have suicidal ideation [37]. In addition, with a sample of 74,698 adolescents in the 2007 KYRWBS, the authors found that, compared to the correct estimation, the overestimation of weight was associated with suicidal ideation in both genders and the underestimation of weight was associated with suicidal attempts only in girls [38]. In this study, our results filled the current research gap in China and showed that the overestimated perception was significantly associated with increased reports of suicidal ideation among girls and attempts among all study adolescents. Nevertheless, these above findings are mainly from cross-sectional analyses, and thus prospective evidence linking weight misperception with suicidal behaviors is warranted.

The strengths of this study included the large sample of Chinese school-based students, a wide adjustment for potential confounders and the examination of the associations between perceived weight status, particularly weight misperception, and suicidal behaviors. Some limitations needed to be taken into account. First, given the cross-sectional nature of the analyses, causality of the reported associations between weight perceptions and suicidal behaviors cannot be inferred from this study. Second, data of height and weight were from self-administered questionnaires and might be subject to reporting bias. According to the previous validation studies, adolescents with self-reported height and weight have been found to underestimate the prevalence of obesity and weight misperception [39,40]. Third, the suicidal behaviors and most of the other variables in the analysis were measured by a single item with dichotomous options, which would inevitably lead to information loss and the bias of the final results. Fourth, as the sample of Chinese students were selected from Zhejiang Province, similar to other local studies, caution is needed in generalizing these findings to the broader Chinese adolescents.

## 5. Conclusions

In conclusion, self-perception of overweight and overestimated perception were associated with increased likelihood of suicidal ideation and attempts in Chinese middle and high students, while the associations with suicidal ideation appeared to be significant only in girls. Carefully designed prospective studies should explore these relationships further. Meanwhile, from the prospective of suicide prevention, teachers, parents and policy makers should pay more attention to adolescents with overweight and overestimated perceptions and further consider the development of tailored interventions based on understanding how social factors (e.g., mass media), lifestyle and weight control behaviors, and mental-health-related characteristics are linked with weight perception.

## Figures and Tables

**Table 1 behavsci-13-00008-t001:** Characteristics of study adolescents in Zhejiang Province according to suicidal ideation, plans and attempts.

Characteristics	Overall	Suicidal Ideation	Suicidal Plans	Suicidal Attempts	None	*p* Value ‡
No. of adolescents	17,359	2833	1088	637	14,254	
Mean age at survey, years	15.7	15.6	15.5	15.4	15.7	<0.001
Boys, %	49.6	39.8	42.7	36.6	51.6	<0.001
Urban school, %	39.1	39.9	37.4	37.1	39.1	<0.001
Middle school level, %	51.3	53.3	56.1	62.3	50.8	<0.001
School performance (below average), %	24.9	32.2	32.5	37.5	23.3	<0.001
Middle school or below (paternal), %	60.2	58.7	58.2	61.6	60.5	0.05
Middle school or below (maternal), %	65.4	64.7	62.3	65.3	65.6	0.04
Lifestyle factors and physical measurement, %						
Current smokers	5.5	9.5	11.9	17.1	4.7	<0.001
Current drinkers	22.3	33.7	40.0	47.8	20.0	<0.001
Muscle-strengthening activity	58.4	54.0	55.4	56.4	59.2	<0.001
Overweight/obesity	8.8	8.6	8.4	8.3	8.9	0.98
Other mental-health-related characteristics, %						
Feelings of loneliness	78.6	93.8	92.7	92.0	75.4	<0.001
Sleep loss due to worry	61.2	79.2	80.0	80.4	57.4	<0.001
Experience of sadness/despair	15.7	38.0	46.2	47.6	11.2	<0.001
Weight perceptions and weight-control-related behaviors, %						
Self-perceived overweight	34.2	41.5	38.9	43.6	32.7	<0.001
Overestimation perception	35.5	42.7	41.8	46.1	34.2	<0.001
Exercising	50.0	57.1	54.8	56.3	48.5	<0.001
Dieting	3.7	9.2	11.5	16.8	2.5	<0.001
Taking laxatives	2.0	4.1	5.6	8.8	1.6	<0.001
Taking diet pills	2.9	6.2	8.2	10.9	2.2	<0.001
Fasting	25.7	39.2	40.0	40.3	22.9	<0.001

‡ For continuous and categorical variables, difference in the characteristics of students with or without any suicidal behavior was compared using Kruskal–Wallis test and linear-by-linear association χ^2^-test, respectively.

**Table 2 behavsci-13-00008-t002:** Adjusted odds ratios (95% CIs) of suicidal ideation, plans and attempts by self-perceived weight status and the accuracy of weight perception.

	Total/Cases	Model 1	Model 2	Model 3	Model 4
Suicidal ideation					
Self-perceived weight status					
About right	7426/1045	1.00	1.00	1.00	1.00
Underweight	3993/612	1.20 (1.07–1.34) *	1.11 (0.98–1.26)	1.05 (0.92–1.20)	1.10 (0.96–1.26)
Overweight	5932/1173	1.42 (1.29–1.56) *	1.45 (1.31–1.62) *	1.38 (1.23–1.54) *	1.22 (1.09–1.37) *
Accuracy of weight perception					
Accurate	9466/1380	1.00	1.00	1.00	1.00
Underestimated	1718/242	1.07 (0.92–1.25)	1.05 (0.90–1.24)	1.00 (0.85–1.19)	1.06 (0.89–1.26)
Overestimated	6167/1208	1.30 (1.18–1.42) *	1.31 (1.19–1.44) *	1.27 (1.15–1.41) *	1.15 (1.03–1.27) *
Suicidal plans					
Self-perceived weight status					
About right	7426/415	1.00	1.00	1.00	1.00
Underweight	3993/250	1.20 (1.01–1.42) *	1.07 (0.88–1.29)	1.01 (0.83–1.23)	1.01 (0.83–1.23)
Overweight	5932/423	1.23 (1.06–1.43) *	1.27 (1.08–1.49) *	1.18 (1.00–1.39) *	1.05 (0.89–1.25)
Accuracy of weight perception					
Accurate	9466/529	1.00	1.00	1.00	1.00
Underestimated	1718/104	1.19 (0.95–1.49)	1.21 (0.96–1.53)	1.16 (0.91–1.48)	1.17 (0.91–1.49)
Overestimated	6167/455	1.29 (1.12–1.48) *	1.28 (1.11–1.48) *	1.24 (1.06–1.44) *	1.13 (0.97–1.32)
Suicidal attempts					
Self-perceived weight status					
About right	7426/227	1.00	1.00	1.00	1.00
Underweight	3993/132	1.20 (0.95–1.52)	1.10 (0.85–1.42)	1.06 (0.82–1.38)	1.01 (0.77–1.32)
Overweight	5932/277	1.47 (1.21–1.78) *	1.53 (1.24–1.88) *	1.46 (1.19–1.80) *	1.32 (1.06–1.64) *
Accuracy of weight perception					
Accurate	9466/280	1.00	1.00	1.00	1.00
Underestimated	1718/63	1.33 (0.98–1.80)	1.28 (0.94–1.75)	1.23 (0.89–1.69)	1.16 (0.83–1.61)
Overestimated	6167/293	1.50 (1.25–1.81) *	1.49 (1.24–1.81) *	1.46 (1.20–1.77) *	1.35 (1.10–1.65) *

Model 1 adjusted for age, gender, location of school, school type, school performance, paternal education and maternal education; model 2 adjusted for model 1 plus health behaviors of smoking, alcohol drinking, physical activity and physical measurement of body mass index; model 3 adjusted for model 2 plus mental-health-related characteristics of feelings of loneliness, sleep loss due to worry and experience of sadness/despair; model 4 adjusted for model 3 plus weight-control-related behaviors of exercising, dieting, taking laxatives, taking diet pills and fasting. * Significant results.

## Data Availability

The datasets generated and/or analyzed during the current study are not publicly available due individual privacy information protection but are available from the corresponding author on reasonable request.

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
