# Peer review of "Relationships between Weight Perceptions and Suicidal Behaviors in Chinese Adolescents: Results from an Ongoing School-Based Survey in Zhejiang Province"

_behavsci, 2022, doi:10.3390/bs13010008_

Round 1
Reviewer 1 Report
The paper explored the associations between weight perceptions and adolescent suicidal ideation, plans, and attempts. Participants were from a provincial-level survey conducted in 2017. The sample size was large. The Xs and Ys were appropriately collected and measured. However, much can still be done to improve the manuscript.
First, in the introduction section, the authors began with the prevalence of obesity and its negative consequences, which were not directly related to the topic in the present study. Indeed, some of the literature reviews in the discussion section could be moved to the introduction so that readers can better understand the studies on the relationship between weight perceptions and adolescent mental health and health behaviors. By doing so, the authors could also be better motivated to outline the limitations in prior research and the gaps the current study aimed to fill.
Second, in the analysis, the authors used logit regression models to estimate the net effect of weight perceptions on suicidal indicators while controlling for a wide range of "confounders" step by step. However, in principle, confounders should pre-exist the X's and Y's concerned and not be confused with mediators that channel link between X and Y. It is obvious the set of so-called confounders in the last model, model 4, were indeed mediators. Thus, the results show that weight perceptions affect suicidal outcomes through changes in loneliness, sleep, etc.
Third, a few minor points. The sample size decreased from 23554 to 17359 during the data preparation process. Which factors contribute the most to the missingness? The stratified analysis, which was dumped in the appendix, was something new to the research domain. The sensitivity analysis was not done with a clear rationale. Why is it sufficient to drop those smoking and drinking participants and reexamine the association between x and y? Any prior work doing so?
Reviewer 2 Report
The authors have done excellent work to demonstrate the relationship between self-perceived weight status and suicide behaviors in Chinese adolescents. It is immensely helpful in the establishment of suicide prevention policy. But there are two questions that need to be explained.
1. In line 117, underweight adolescents who perceived themselves as about the right weight were allocated into overestimated perception, it may increase selection bias that will exaggerate the effects of overestimated perception on suicide behaviors. It would be better to exclude this cohort from analysis.
2. There are tremendous confounding factors associated with suicide behaviors, why didn't the authors employ propensity score match method to balance those confounders between groups?
Reviewer 3 Report
Relationships between weight perceptions and suicidal behaviors in Chinese adolescents: results from an ongoing school- 3 based survey in Zhejiang Province
The manuscript focuses on one of the most severe public health current concerns - the Obesity in children and adolescents. It explores its relationship with suicidal behaviors, which is another serious public health problem among adolescents. Considering this research topic, I think it fits perfectly with the objectives of this journal.
Besides the relevant main topics explored in the study, one of the strengths of the study is the sample and age range (17,359 middle and high school students aged 9 to 18 years)
Overall, the article is well-organized, containing all standard components; Introduction, Methods, Results, Discussion and Conclusion.
Title: The title is appropriate.
Abstract: The abstract is very well and includes the most relevant information of the work.
Introduction:
The introduction is clear and centred on the essential points. Below are some comments and suggestions to help the authors complement the text and synthesize the literature in the field.
-Line 49-50: Emphasize that a phenomenon exists in diverse cultures and countries. The authors can include other important studies and countries related to this topic: namely Korean (Kim DK, Song HJ, Lee EK, Kwon JW. Effect of sex and age on the association between suicidal behaviour and obesity in Korean adults: a cross-sectional nationwide study. BMJ Open. 2016 Jun 2;6(6):e010183. doi:10.1136/bmjopen-2015-010183. PMID: 27256086; PMCID: PMC4893869.), China (Guo L, Xu Y, Huang G, Gao X, Deng X, Luo M, Xi C, Zhang WH, Lu C. Association between body weight status and suicidal ideation among Chinese adolescents: the moderating role of the child's sex. Soc Psychiatry Psychiatr Epidemiol. 2019 Jul;54(7):823-833.
doi: 10.1007/s00127-019-01661-6. Epub 2019 Feb 2. PMID: 30712066.); Jeong SC, Kim JY, Choi MH, Lee JS, Lee JH, Kim CW, Jo SH, Kim SH. Identification of influencing factors for suicidal ideation and suicide attempts among adolescents: 11-year national data analysis for 788,411 participants. Psychiatry Res. 2020 Sep;291:113228. doi: 10.1016/j.psychres.2020.113228. Epub 2020 Jun 15. PMID: 32562930); Latin American ( Elia C, Karamanos A, Dregan A, O'Keeffe M, Wolfe I, Sandall J, Morgan C,Cruickshank JK, Gobin R, Wilks R, Harding S. Association of macro-level determinants with adolescent overweight and suicidal ideation with planning: Across-sectional study of 21 Latin American and Caribbean Countries. PLoS Med.2020 Dec 29;17(12):e1003443. doi: 10.1371/journal.pmed.1003443. PMID: 33373361;
PMCID: PMC7771665) and study of Campisi SC, Carducci B, Akseer N, Zasowski C, Szatmari P, Bhutta ZA. Suicidal behaviours among adolescents from 90 countries: a pooled analysis of the global school-based student health survey. BMC Public Health. 2020 Aug 10;20(1):1102.
doi: 10.1186/s12889-020-09209-z. PMID: 32772922; PMCID: PMC7416394.
-Line 50: Before mentioning the impact of overweight or obesity on suicidal behavioral, I think it is important to mention the influential factors of these behaviors (e.g. see the study of Jeong SC, Kim JY, Choi MH, Lee JS, Lee JH, Kim CW, Jo SH, Kim SH. Identification of influencing factors for suicidal ideation and suicide attempts among adolescents: 11-year national data analysis for 788,411 participants. Psychiatry Res. 2020 Sep;291:113228.
doi: 10.1016/j.psychres.2020.113228. Epub 2020 Jun 15. PMID: 32562930)
-Line 54: specify what specific papers refer to these factors and add others;
-e.g. such gender and age (study of Kim DK, Song HJ, Lee EK, Kwon JW. Effect of sex and age on the association between suicidal behaviour and obesity in Korean adults: a cross-sectional nationwide study. BMJ Open. 2016 Jun 2;6(6):e010183. doi: 10.1136/bmjopen-2015-010183. PMID: 27256086; PMCID: PMC4893869 and Guo, L., Xu, Y., Huang, G., Gao, X., Deng, X., Luo, M., Xi, C., Zhang, W. H., Lu, C. Association between body weight status and 330 suicidal ideation among Chinese adolescents: the moderating role of the child's sex. Soc Psychiatry Psychiatr Epidemiol 2019, 54, 331 823-833. doi: 10.1007/s00127-019-01661-6.
- victimization: van Vuuren, C. L., Wachter, G. G., Veenstra, R., Rijnhart, J., van der Wal, M. F., Chinapaw, M., Busch, V. Associations between 327 overweight and mental health problems among adolescents, and the mediating role of victimization. BMC Public Health 2019, 328
19, 612. doi: 10.1186/s12889-019-6832-z
-And add others, like race and ethnicity:
Erausquin JT, McCoy TP, Bartlett R, Park E. Trajectories of Suicide Ideation
and Attempts from Early Adolescence to Mid-Adulthood: Associations with
Race/Ethnicity. J Youth Adolesc. 2019 Sep;48(9):1796-1805. doi:
10.1007/s10964-019-01074-3. Epub 2019 Jul 12. PMID: 31301028.
- At the end of the introduction and after the main objectives of the study, which are the study's main hypotheses based on the literature review?
Concerning Method:
Measures: It is essential to discuss the reliability of the measures used ("yes" or "no" response options seem to be very restricted to assessing suicidal behaviors and lifestyle variables).
I think there is a lack of a good reflection on how these variables were evaluated and the possible impact of these limitations on the results.
On the other hand, to what extent does a child know and identify what is suicidal ideation or a suicide attempt or suicidal plans? These reflections are important in the discussion.
Statistical procedure – Although the statistical procedures used are adequate to the research questions, a key issue for me is that considering the aim of the study and the different levels of influence of the confounder variables (contextual, like school variables, parental educational variables, individual variables, age, gender, etc), why didn´t the authors considered a multilevel binary logistic regression?
Results:
Line 164: Provide Mean and SD values of sample´s age
Line 174-179: You must provide satistical evidence for these affirmations. What statistical tests did the authors use to do these conclusions? (and put which tests used in the point above - statistical procedure)
Results are well presented, and the complementary material is relevant to point the main objectives of the study
Discussion:
I think there are some reflections (some I suggested above) that will be important to enrich and deepen the discussion.
- The way of assessing the variables
- Comment on the results: Suicidal ideation (n=2,833); Suicidal plans (n=1,088) and Suicidal attempts (n=637). What characteristics does this Province have that could justify some of these results? How do these results fit in with other areas of the country or even other countries?
- The longitudinal study mentioned in line 227 had the same measures and variables that this one?
Conclusion:
Considering prospective of suicide prevention and involving diverse and systemic variables with different levels of analyze, I think the prevention, besides the role of parents and teachers, must also consider socio-political changes, with attention to social factors such as social media and the impact of social networks on the behavior and experiences of children and adolescents. Consider the development of tailored and ecological interventions.
